# Innovative strategies and implementation science approaches for health delivery among migrants in humanitarian settings: A scoping review

**Christopher W. Reynolds**[1,2]*, **Jennifer Y. Rha**[3], **Allison M. Lenselink**[2,4], **Dhanya Asokumar**[1], **Laura Zebib**[1], **Gurpreet K. Rana**[5], **Francesca L. Giacona**[1,2], **Nowshin N. Islam**[6], **Sanjana Kannikeswaran**[1], **Kara Manuel**[1], **Allison W. Cheung**[1], **Maedeh Marzoughi**[1], **Michele Heisler**[7,8]

1 Department of Surgery, University of Michigan Medicine, Ann Arbor, Michigan, United States of America, 2 Physicians for Human Rights Student Advisory Board, New York, New York, United States of America, 3 Rutgers Robert Wood Johnson Medical School, Newark, New Jersey, United States of America, 4 Medical School for International Health at Ben Gurion University, Beer Sheva, Israel, 5 Taubman Health Sciences Library, University of Michigan, Ann Arbor, Michigan, United States of America, 6 CUNY School of Medicine, New York, New York, United States of America, 7 Department of Internal Medicine, University of Michigan Medical School, Ann Arbor, Michigan, United States of America, 8 Department of Health Behavior and Health Equity, University of Michigan School of Public Health, Ann Arbor, Michigan, United States of America

* chwre@med.umich.edu

## Abstract

### Introduction

Over 100 million displaced people rely on health services in humanitarian contexts, defined as unstable or transitory settings created in response to complex emergencies. While services are often described, there is a dearth of evidence on best practices for successful implementation to guide efforts to optimize health delivery. Implementation science is a promising but underutilized tool to address this gap. This scoping review evaluates implementation science in health services for forced migrants in humanitarian settings.

### Methods

We conducted a scoping review according to JBI methodologies. A search of eight databases yielded 7,795 articles, after removal of duplicates, that were screened using PRISMA-ScR guidelines. Data extraction assessed study descriptors, implementation objects, barriers, facilitators, implementation strategies, and use of implementation frameworks in service delivery.

### Results

Data from 116 studies represented 37 countries and 11 topic areas. Methods were mainly cross-sectional with low-medium evidence rigor. Mental health programs (25%) and vaccination services (16%) were the most common objects of implementation. Thirty-eight unique barriers were identified including resource limitations (30%), health worker shortages

**Data Availability Statement:** The literature search strategy and full-list of citations can be found in the

data repository available from: https://dx.doi.org/10.7302/22500.

**Funding:** The authors received no specific funding for this work.

**Competing interests:** The authors have declared that no competing interests exist.

(24%), and security risks (24%). Among 29 facilitators, the most common were health worker availability (25%), pre-existing partnerships (25%), and positive perceptions towards the intervention (20%). More than 90% of studies collectively identified 35 implementation strategies, the most common being capacity building (44%), stakeholder engagement (35%), information dissemination (38%), and feedback mechanisms (25%). Only 10 studies used formal implementation models, with RE-AIM (n = 3) and Intervention mapping (n = 2) being most frequent.

## Conclusions

In this scoping review, we found similar barriers, facilitators, and implementation strategies across diverse humanitarian migrant settings and services. However, the use of rigorous methods and formal implementation models was rare. Frameworks included RE-AIM, CFIR, and Precede-Proceed. Increased use of implementation science frameworks and methods will help humanitarians more rigorously and systematically evaluate and develop best practices for implementation of health services for migrants in humanitarian settings.

## Introduction

With an increase in complex emergencies, over 100 million people are forcibly displaced worldwide [1]. Many reside within refugee camps, congregate settlements, or transitory housing in urban settings. Displaced persons' access to essential health care is often hindered due to multiple factors, including limited funds or insurance, a lack of knowledge, and restrictive host country policies which limit integration into formal health systems [2]. At the same time, migrants often have increased health needs, including physical and mental sequelae which are secondary to stressors before and during migration [3]. Today, the average length of migrant displacement is 17 years, indicating a need for chronic, longitudinal care systems [4]. In place of formal systems, migrant health services are often provided through a complex web of local and international actors, non-government organizations (NGOs), and government partnerships [5]. United Nations High Commissioner for Refugees, Médecins Sans Frontières, and International Committee of the Red Cross are some of the most well-known organizations, but more than 11,000 exist today in the United States alone [6].

Efforts to optimize migrant health have focused overwhelmingly on improving care access once resettled in high-income host countries [7]. However, the majority of refugees remain displaced in low- and middle- income countries (LMIC), and there is a paucity of data on how care is effectively delivered within transitional spaces prior to being resettled [8]. Many argue that ineffective care in complex emergencies can be attributed to a lack of culturally appropriate interventions, inadequate coordination, and financial constraints [9]. With the chronicity of protracted humanitarian events and increased difficulties to access pathways for resettlement, the number of individuals within transitional settings is likely to rise. Therefore, it is imperative to understand best practices for delivering acute and chronic care in such transitional humanitarian settings to prevent morbidity and mortality.

In a system of limited resources, effective implementation of interventions and practice changes is critical to ensure optimized and ethical care delivery [10]. Implementation science is the transdisciplinary study of methods and strategies that promote the adoption and integration of scientific evidence, implementation of interventions, and efforts to sustain and scale

them [11]. Implementation outcomes are distinct from clinical outcomes and focus on the adoption of evidence-based practice, appropriateness of the evidence for local context, and cost, feasibility, and sustainability of interventions. Implementation science seeks to bridge the research-to-practice gap and has been effectively utilized within high-income country (HIC) health systems to optimize service delivery and enhance health outcomes [12, 13].

Within global health, implementation science is becoming recognized as a cornerstone to improve equity by tailoring strategies to complex social needs and ensure scalability of proven approaches [14, 15]. However, there has been little emphasis on using implementation science to improve care delivery in humanitarian settings. Within refugee resettlement, implementation science has highlighted barriers to implementation that may be universal and impact healthcare implementation such as lack of time, budget constraints, and language barriers [16]. One scoping review of conflict zones found that implementation experts are often engaged after research is already conducted, and that most humanitarian organizations lack expertise in technical research skills [17]. The very nature of humanitarian crises, including socio-political instability, population mobility, and limited infrastructure creates research challenges for the proper use of implementation science and its principles [18].

To align with growing practices encouraging collection of empirical results, many NGOs and governments have adopted methods to evaluate their programs objectively [19, 20]. Specifically, process evaluations offer a way to understand the dynamic interplay among human factors, local context, and material resources that facilitate program success [21]. However, it is unknown whether, and if so how, specific implementation science approaches and frameworks are utilized to investigate factors influencing program outcomes in health care service delivery for migrants in transitional, humanitarian settings.

Accordingly, the purpose of this scoping review was to understand the extent and type of evidence for effective implementation of health care to migrant populations in humanitarian settings. Specifically, we aimed to determine what programs are being implemented, the rigor with which objects are being studied, which barriers and facilitators exist to program implementation, the availability and use of evidence-based implementation strategies, and the extent to which formal implementation theories, models, or frameworks were used for designing and evaluating health programs. First, we sought to identify and categorize objects of implementation: the health interventions delivered to migrants in transitory settings. Second, we describe the determinants, barriers and facilitators, identified by study authors. Lastly, we describe utilization of implementation strategies and formal theories, models, and frameworks for study design. This review seeks to comprehensively identify the key uses and existing gaps in the use of implementation science for care delivery among humanitarian migrant populations. Improving understanding of effective implementation within humanitarian settings is essential to improving health outcomes and equity for this vulnerable population.

## Materials and methods

This study utilized the JBI Manual for Evidence Synthesis for Scoping Reviews methodology [22]. JBI presents a comprehensive framework for conducting scoping reviews and covers why a scoping review should be conducted, how to develop a protocol, and the search strategies, data extraction, and results presentation to do so with a rigorous and validated approach.

### Review questions

The following study questions guided our review:

1. What objects of implementation are being studied among migrants in humanitarian settings?

2. Which determinants (barriers and facilitators) are described regarding program implementation among migrants in humanitarian settings?

3. Which implementation strategies are being used to implement programs among migrants in humanitarian contexts?

4. How often and which implementation theories, models, and frameworks are being used to guide humanitarian research among migrant populations?

## Search strategy

A search of the literature was conducted by a health sciences informationist (GKR) in October 2022 and a search update was implemented in February 2024. Eight scholarly databases were searched: MEDLINE (via Ovid interface), EMBASE (via Embase.com), CINAHL (via EBSCO-host), Scopus, PsycINFO (via EBSCOhost), Web of Science Core Collection (via Thomson Reuters), CENTRAL (Cochrane Central Register of Controlled Trials) in Cochrane Library (via Wiley) and the Cochrane Covid-19 Study Register (via Wiley). Keywords and controlled vocabulary search terms were used to represent the three main search concepts: 1) displaced persons; 2) humanitarian settings: and 3) health care delivery or disease management. The three main search concepts were combined to develop the final search strategies. Test searching was used to determine variation in controlled vocabulary terminology and search syntax. A revised version of University of Alberta's refugee camps search hedge for Ovid MEDLINE was utilized in each database search [23]. Search results were limited to English or Spanish articles published from 2000 to February 2024 in six of the eight databases. Language limits were not used in the search of the CENTRAL; and language or year limits were not required for the search implemented in the Cochrane Covid-19 Study Register.

A total number of 10,857 citations were exported to the citation manager EndNote (Clarivate Analytics) for processing and removal of duplicate articles. After removal of duplicates using a variation on the Bramer method [24], 7,795 citations were exported to the evidence synthesis screening tool Rayyan [25] for assessment and initial screening. Due to the comprehensive search strategies implemented, keywords needed in the search strategy, and complexity of displaced person status, it was not always possible to clearly differentiate between articles addressing displaced persons in humanitarian settings from those with resettlement experiences during the search process. As such, a higher proportion of literature not relevant to this review was retrieved. Articles addressing resettlement experiences were removed in the initial screening. The literature search strategies and citation files are available at https://dx.doi.org/10.7302/22500 [26].

## Inclusion criteria

For this review, we selected research which met the following inclusion criteria:

1. Studies of migrant persons, including refugees, asylum seekers, and internally displaced persons (IDPs) of all ages who have received physical or mental health care services in a humanitarian setting.

2. Studies that were conducted in a humanitarian setting, defined as an unstable, transitory, or externally-supported context or infrastructure created in response to a complex

emergency or crisis. Examples included refugee or IDP camps, United Nations or governmental transitional spaces, and congregate settlements.

3. Studies that described the implementation planning, process, and/or outcomes of a health service provided in these settings.

### Exclusion criteria

We excluded studies deemed to be not sufficiently within the inclusion criteria. Specific exclusion points were studies on migrants who had been resettled in a host country and were no longer in a humanitarian setting, observational or cross-sectional studies which solely described health status or outcomes without descriptions of implementing a health service, and studies that postulated theoretical health services or presented pre-implementation acceptability data without an actual experience of implementing care. Studies which were not published in English or Spanish were excluded from this scoping review. Given the nascency of implementation science in humanitarian research, we did not exclude studies that failed to explicitly define determinants, implementation strategies, and frameworks with academic terms. Instead, this information was extracted by hand during the review process by researchers familiar with implementation literature. In some cases, implementation information was not the main focus of the study but was described sufficiently in the methods. Such articles were still included. We also did not exclude studies which failed to report on health outcomes, as our primary objective was to evaluate implementation processes. Editorials, letters to the editor, and opinion articles which did not report on original research were excluded.

### Article screening and data extraction

Following the database search, we screened article titles and abstracts for alignment with the study inclusion criteria. First, a team of researchers was assembled and trained on article screening in accordance with our study objectives. This process involved two authors randomly selecting 25 articles and independently labeling them as 'include' or 'exclude'. These researchers met and discussed any discrepancies to reach consensus on the 25 pilot articles. As a method to train the complete study team, all members involved in screening first underwent a study orientation and independently screened the same select 25 articles. Their answers were then compared to the consensus decision and each individual met with the lead authors to identify discrepancies in their screening decisions in order to clarify accurate procedures. This process was continued until each individual screener reached at least an 80% consensus decision.

The formal article screening followed PRISMA-ScR (Preferred Reporting Items for Systematic Reviews and Meta-Analyses-Scoping Review extension) guidelines [27]. We began first with an abstract and title review of the 7,795 retrieved articles (Fig 1). Each article title and abstract was reviewed independently by at least two researchers. When screening decisions between researchers agreed, the article was either excluded or passed to the full-text screening phase. If there was disagreement, a third, more senior team member resolved the conflict with a final decision. Following the abstract screen, we conducted a full-text screen of the remaining 394 articles. This phase used the same process as the initial screening, utilizing at least two researchers with conflict resolution by a third researcher when necessary [28].

Data extraction included study titles, authors, year, location by country, methods type, topic area, key findings from the study, and descriptions of determinants (barriers and facilitators), implementation strategies, and formal implementation science theories, models, or frameworks. We also determined if each article reported on the scope of care delivered (i.e.

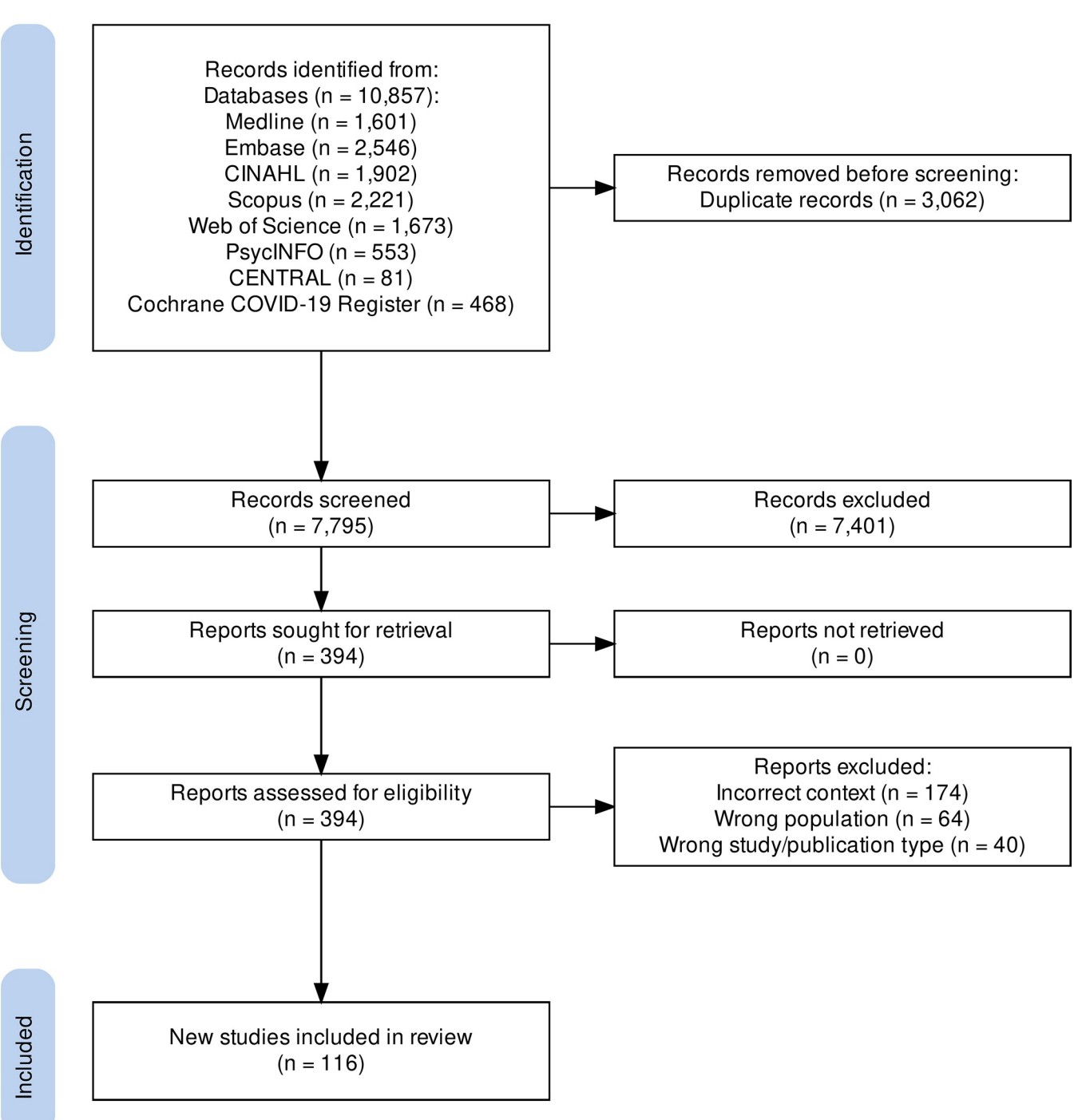

**Fig 1. PRISMA flowchart describing study screening, exclusion, and selection.** Diagram describing study selection and inclusion beginning with the database searches and following the Preferred Reporting Items for Systematic Reviews and Meta-Analyses extension for Scoping Reviews (PRISMA-ScR) to screen abstracts and articles.

how many patients the service reached) and if health outcomes such as clinical effectiveness data were studied. We classified the rigor of each study from one to seven according to the Melnyk and Fineout-Overholt evidence hierarchy classification [29]. Results were organized primarily by topic area then alphabetically by study title.

### Ethics approval statement

This study does not involve human participants so Institutional Review Board review was not applicable.

## Results

### Study characteristics

This review included 116 studies that met inclusion criteria. Studies were situated within a range of humanitarian contexts across five continents. Most occurred in the Middle East and Sub-Saharan Africa, followed by Asia. The Americas were the least represented regions (5%). Thirty-seven countries were included, with Bangladesh (18%), Uganda (12%), and Lebanon (9%) with the most studies. We categorized implemented programs into 11 overarching health topic areas. The most frequent were infectious disease (31%), mental health (26%), maternal and child health (12%), preventive or non-communicable disease care (10%), and health technology (9%). Less common were sexual and reproductive health (6%), ophthalmology (2%), surgery (2%), pharmacy (1%), and nutrition (1%) (Table 1, S1 Appendix).

Study rigor and methodologies varied from weak to strong evidence. The most common methodologic approach was a single-site case study (55%), many of which used descriptive, cross-sectional, and qualitative methods. For example, Murphy et al. used a qualitative nested design to determine that fixed-dose combination therapy was acceptable and feasible for cardiovascular disease prevention among Syrian refugees in Lebanon [30]. The second most frequent method was quasi-experimental implementation cohort studies, which regularly utilized pre- and post-evaluation measures to evaluate program impact within a migrant population (16%). Amsalu et al. used a pre- and post-descriptive study to evaluate a neonatal survival program across refugee camps in Somalia, Niger, Chad, and Cameroon [31], while Phares measured the impact of an oral cholera vaccination campaign among 35,000 refugees in Thailand [32]. Only ten studies (9%) were randomized control trials (RCTs) [33–38]. Randomization occurred at the patient or cluster level most commonly across sections of refugee camps. Trials that were randomized at the cluster level justified their approach through ethical concerns of randomizing individuals to humanitarian interventions considered basic rights [33]. Use of more complex methods for multi-intervention testing was rare, as was the case with a 2 x 2 factorial design to evaluate the impacts of conditional cash transfers and an mHealth program on vaccine coverage and malnutrition in Somalia [39]. Four reviews, one scoping [40] and three systematic [41–43], were included, but none were level 1 rigor representing a systematic review of RCTs.

Studies were analyzed for reach (i.e. number of patients impacted) and clinical outcomes (i.e. change in disease state). While 99 studies reported reach (85%), only 37 provided clinical outcomes data (32%). Generally, studies with stronger evidence ratings of 4 and above were more likely to report clinical outcomes. For example, Lawrence et al. quantified a significant decrease in psychological dysfunction among IDPs in Nigeria through a quasi-experimental factorial design using randomized cohort trial and cluster sampling [44]. By summing information extracted on the scope of each study, we determined that at least 4,991,937 migrants were impacted by the implemented programs in this review.

**Table 1. Descriptives, key findings, and implementation outcomes for studies on migrant health services in humanitarian settings.**

| Citation | Country | Evidence level | Study type | Object of implementation and findings | Determinants | Implementation strategies | Theories, models, or frameworks | Scope | Clinical outcomes |
|---|---|---|---|---|---|---|---|---|---|
| | | | | Infectious disease (n = 36, 31%) | | | | | |
| Abdullahi et al. (2020) | Nigeria | 3 | Pre/post intervention study with interrupted time series analysis | ID screening; a TB case finding intervention led to 283,556 screenings across 26 IDP camps | X | X | | X | X |
| Ali et al. (2022) | Sudan | 6 | Case study | ID screening; intervention increased, training, coordination, and follow-up to limit COVID-19 transmission in South Korfodon, Sudan | X | X | | X | |
| Amani et al. (2021) [82] | Cameroon | 6 | Cross-sectional | Vaccination campaign; preventive mass vaccination against meningococcal meningitis was successful | X | X | | X | |
| Beshr et al. (2023) | Yemen | 4 | Case study | Vaccination campaign; 7.2 million children vaccinated for polio and multiple sanitary measures established in IDP camps in conflict-affected areas | X | X | | X | X |
| Center for Disease Control (2004) | Sudan | 6 | Case study | Vaccination campaign; mass vaccination in Darfur IDP camps successfully reduced measles cases | X | X | | X | X |
| Chowdury et al. (2023) [75] | Bangladesh | 6 | Case study | Vaccination campaign; COVID-19 campaign among Rohingya exposed determinants to project success | X | X | | | |
| Coldiron et al. (2017) | Uganda | 3 | Quasi-experimental cohort study | ID treatment; intermittent preventive malaria treatment reduced incidence among 40,611 refugees | X | X | | X | X |
| Donkeng-Donfack et al. (2022) | Cameroon | 6 | Cost-benefit analysis | ID screening; TB-LAMP as an initial diagnostic followed by Xpert MTB/BRIF for TB screening in mass campaigns has a cost benefit after 12 months of use | X | | | X | |
| Eisenberg et al. (2021) [81] | Bangladesh | 6 | Retrospective cohort | ID treatment; DAT was feasible and safe among 709 patients in a low-resource environment with appropriate monitoring, training, and safety awareness | X | X | | X | X |
| Ghanwji et al. (2023) | Syria | 6 | Case study | ID screening; 24,956 participants COVID-19 antigen tested by community health workers | X | X | | X | |

*(Continued)*

**Table 1.** (Continued)

| Citation | Country | Evidence level | Study type | Object of implementation and findings | Determinants | Implementation strategies | Theories, models, or frameworks | Scope | Clinical outcomes |
|---|---|---|---|---|---|---|---|---|---|
| Grijalva-Eternod et al. (2023) [39] | Somalia | 2 | 2x2 factorial cluster-randomized controlled trial | Vaccination; 1,430 households randomized to cash transfer and an mHealth intervention showing cash transfer improved measles and pentavalent vaccine coverage | X | X | | X | X |
| Halder et al. (2023) | Bangladesh | 6 | Case study | ID prevention; risk communication and community engagement facilitated COVID-19 prevention measures | X | X | | | |
| Kaic et al. (2001) | Croatia | 4 | Cohort implementation study | ID screening; prospective testing and immunization of seronegative patients was more cost effective and adequately controlled a hepatitis A outbreak in a Croatian refugee camp | X | X | | X | X |
| Khan et al. (2023) | Bangladesh | 6 | Case study | Vaccination campaign; Seven rounds of oral cholera vaccine implemented among 900,000 Rohingya, resulting in no subsequent outbreaks | X | X | | X | X |
| Knust et al. (2022) | Thailand | 6 | Case study | ID screening; surveillance testing system implemented, accounting for 6,190 specimens | X | X | | X | |
| Koop et al. (2001) | North Macedonia | 6 | Case study | Vaccination campaign; implementing an expanded immunization program without a stable population was problematic | X | X | | X | |
| Korave et al. (2021) | Nigeria | 6 | Case study | Vaccination campaigns; suboptimal coverage may be due to unstable IDP population and routine immunizations is essential to herd immunity | X | X | | X | |
| Kouadio et al. (2009) | Ivory Coast | 6 | Cross-sectional | ID screening; disease surveillance teams for measles and rubella controlled outbreaks through surveillance and serological testing | X | X | | X | |
| Lam et al. (2017) | Iraq | 6 | Cross-sectional | Vaccination campaigns; oral cholera vaccine was successfully implemented among a high-risk, conflict-affected population | X | X | | X | |

*(Continued)*

**Table 1.** (Continued)

| Citation | Country | Evidence level | Study type | Object of implementation and findings | Determinants | Implementation strategies | Theories, models, or frameworks | Scope | Clinical outcomes |
|---|---|---|---|---|---|---|---|---|---|
| Mellou et al. (2019) | Greece | 6 | Cross-sectional | Vaccination campaign; vaccine intervention led to 57,615 doses and high coverage for MMR but not for other vaccines | X | X | | X | |
| O'Laughlin et al. (2014) | Uganda | 3 | Cohort implementation study | ID screening; clinic-based HIV testing in a refugee settlement was feasible and led to 6x increase in positive cases | | | | X | |
| Oladeji et al. (2019) [90] | South Sudan | 4 | Cohort implementation study | Vaccine services; integrating immunization with health services increased distribution and decreased dropout rate | X | X | X | X | |
| Ope et al. (2017) | Kenya | 6 | Field function study | ID screening; ICS-RV rotavirus diagnostic test was feasible in a Kenyan refugee camp | X | X | | X | |
| Phares et al. (2016) [32] | Thailand | 4 | Cohort implementation study | Vaccination campaign; 63,057 oral cholera vaccine doses were administered to 35,399 refugees with low adverse effects | | X | | X | X |
| Polonsky et al. (2021) | Bangladesh | 4 | Retrospective review | Vaccination campaign; 89% of individuals vaccinated for diphtheria following an outbreak in Cox's Bazar | | X | | X | |
| Porta et al. (2014) [73] | South Sudan | 4 | Cohort implementation study | Vaccination campaign; Using two strategies for vaccination implementation showed feasibility and high coverage capabilities | X | X | | X | |
| Rabiou et al. (2022) | Niger | 6 | Case study | ID screening; COVID-19 surveillance screened 169,475 migrants for fever | X | X | | X | |
| Rainey et al. (2013) [68] | Haiti | 4 | Cohort implementation study | Vaccination campaign; rapid monitoring of vaccination is only marginally beneficial to reach immunization targets in temporary camps | X | X | | X | |
| Ravicz et al. (2022) [51] | Uganda | 6 | Qualitative cross-sectional | ID treatment; stakeholders and community members co-designed an HIV linkage program | X | X | X | | |
| Rutta et al. (2008) [76] | Tanzania | 4 | Cohort implementation study | ID treatment; PMTCT was successfully integrated into antenatal care and acceptable to most pregnant women | X | X | | X | |

(*Continued*)

**Table 1.** (Continued)

| Citation | Country | Evidence level | Study type | Object of implementation and findings | Determinants | Implementation strategies | Theories, models, or frameworks | Scope | Clinical outcomes |
|---|---|---|---|---|---|---|---|---|---|
| Seal et al. (2023) | Somalia | 2 | Randomized controlled trial | Vaccination program; adapted Participatory Learning and Action improved measles and pentavalent vaccine coverage among 1,520 IDPs | X | X | | X | X |
| Sheikh et al. (2014) [46] | Kenya | 4 | Cohort implementation study | Vaccination campaign; Polio vaccine distribution to 2,000 children by 299 teams | X | X | | X | |
| Stein et al. (2022) | Uganda | 3 | Quasi-experimental mixed methods | Infection prevention; Households with cash transfers were more food and psychologically secure but had no impact on COVID-19 preventive measures | X | | | X | X |
| Van Boetzel et al. (2020) | Bangladesh | 6 | Case study | ID screening; 97,340 households participated in community-based surveillance for epidemic disease | X | X | | X | |
| Varkey et al. (2009) | India | 6 | Case study | Vaccination campaign; measles vaccine for children 6 months to 14 years prevented outbreaks and death in camps | X | X | | X | |
| White et al. (2019) | Ethiopia | 6 | Cross-sectional | The supertowel was an acceptable complement to soap among refugees in a camp setting | X | X | | | |
| Mental health (n = 30, 26%) | | | | | | | | | |
| Akhtar et al. (2021) [71] | Jordan | 4 | Feasibility randomized control trial | Mental health program; GroupPM+ was feasible, culturally acceptable, and decreased psychological distress for 64 Syrian refugees | X | X | | X | X |
| Bastin et al. (2013) [77] | Lebanon | 6 | Retrospective cohort | Mental health program; improved functionality among 1144 patients with needs to adapt to more severe cases | X | X | | X | X |
| Blackwell et al. (2022) [72] | Iraq | 4 | Cohort implementation study | Mental health program; local psychotherapists reduced PTSD and depression scores among 28 IDPs | X | X | | X | X |
| Bolton et al. (2007) [38] | Uganda | 2 | Randomized controlled trial | Mental health program; group psychotherapy improved depression symptoms in refugee girls but not boys among 314 participants | X | X | | X | X |

(*Continued*)

**Table 1.** (Continued)

| Citation | Country | Evidence level | Study type | Object of implementation and findings | Determinants | Implementation strategies | Theories, models, or frameworks | Scope | Clinical outcomes |
|---|---|---|---|---|---|---|---|---|---|
| Borja et al. (2019) [88] | Bangladesh | 6 | Case study | Mental health program; psychosocial services should be consistent and prioritized by local agencies | X | | X | | |
| Bosqui et al. (2023) | Lebanon | 3 | Multiple n = 1 implementation design | Mental health program; telephone-delivered mental health program showed increased impact among Syrian refugees with war-related trauma | X | X | | X | X |
| Bryant et al. (2022) [34] | Jordan | 2 | Randomized control trial | Mental health program; lay-provider group therapy yielded positive results for depressive symptoms, self-awareness, and parenting skills among 410 refugees | X | X | | X | X |
| Bryant et al. (2022) [35] | Jordan | 2 | Randomized control trial | Mental health program; Syrian refugees aged 10–14 showed reduced problem internalization following EASE intervention | X | X | | X | X |
| Bryant et al. (2022) | Jordan | 2 | Randomized control trial | Mental health program; gPM+ showed depression reductions at 3 months but not 1 year | X | X | | X | X |
| Cohen et al. (2021) [40] | Multi-site | 5 | Scoping review | Mental health program; RE-AIM assessed adaption, feasibility, and fidelity to determine a dearth of capacity building for lay providers in psychosocial humanitarian efforts | X | X | X | X | |
| Corna et al. (2019) | Bangladesh | 6 | Case study | Mental health program; psychosocial intervention was effective among 260 pregnant women | X | X | | X | X |
| Crombach et al. (2018) | Burundi | 3 | Cohort implementation study | Mental health program; NET therapy improved PTSD and depression symptoms among 51 participants | X | X | | X | X |
| Dyer et al. (2019) | Bangladesh | 6 | Case study | Mental health program; psychosocial program implemented among 664 Rohingya refugees in Cox's Bazar for several mental health disorders | X | X | | X | |
| Ezard et al. (2010) [56] | Thailand | 6 | Case study | Substance use screening; considered feasible among 1,256 males screened for high-risk alcohol use | X | X | | X | |

(*Continued*)

**Table 1.** (Continued)

| Citation | Country | Evidence level | Study type | Object of implementation and findings | Determinants | Implementation strategies | Theories, models, or frameworks | Scope | Clinical outcomes |
|---|---|---|---|---|---|---|---|---|---|
| Greene et al. (2022) | Tanzania | 6 | Feasibility cluster randomized trial | Mental health program; a multi-sectoral violence and mental health-focused intervention was beneficial but required adaptation | X | X | | | |
| Greene et al. (2021) [36] | Tanzania | 2 | Cross-sectional, qualitative | Mental health program; Nguvu run by lay refugee workers reduced psychological stress for victims of interpersonal violence among 311 Congolese women but did not reduce violence itself | X | X | | X | X |
| Keshk et al. (2021) | Anonymized | 6 | Retrospective cohort study | Mental health program; clinic provided multidisciplinary services to migrants with varied rates of mental, physiotherapy, and social care | X | X | | X | |
| Lawrence et al. (2020) [44] | Nigeria | 3 | Quasi-experimental factorial design with randomized cohort trial and cluster sampling | Mental health program; trauma-focused counseling and social effectiveness skills significantly improve functioning of displaced adolescents | | | | X | X |
| Levy et al. (2020) | Serbia | 6 | Case study | Mental health program; six C's method was feasible to provide psychosocial support in Presevo refugee camp | X | X | | | |
| Mahmuda et al. (2019) [45] | Bangladesh | 6 | Case study | Mental health program; psychotherapy intervention with group adapt therapy was successfully implemented by 17 counselors among 116 clients | X | X | | X | |
| Mercer et al. (2005) | India | 6 | Cross-sectional | Mental health program; spiritual and cultural practices were most important in delivering mental health services | X | X | | | |
| Metzler et al. (2023) | Uganda | 2 | Randomized controlled trial | Mental health program; Toolkit for Friendly Spaces in Humanitarian Settings and standard intervention were successful first-line for 1,280 adolescents | X | X | | X | X |
| Momotaz et al. (2019) | Bangladesh | 6 | Cohort implementation study | Mental health program; positive increase in health worker provision of mental health care | X | X | | X | |

(*Continued*)

**Table 1.** (Continued)

| Citation | Country | Evidence level | Study type | Object of implementation and findings | Determinants | Implementation strategies | Theories, models, or frameworks | Scope | Clinical outcomes |
|---|---|---|---|---|---|---|---|---|---|
| Murray et al. (2018) [89] | Ethiopia, Somalia | 4 | Quasi-experimental | Mental health program; CETA significantly reduces PTSD symptoms and increases well-being in refugee youth and caregivers | X | X | X | X | X |
| Quosh (2016) | Syria | 6 | Case study | Mental health referrals; case management can support uptake of mental health services among 6,000 clients | X | X | | X | |
| Saeed Khan et al. (2022) [78] | Qatar | 6 | Case study | Mental health referrals; 30 refugee children successfully referred to mental health services | X | X | | X | |
| Sonderegger et al. (2011) [68] | Uganda | 4 | Cohort non-randomized control trial | Mental health program; EMPOWER CBT reduced depression and anxiety among IDPs | | X | | X | X |
| Sullivan et al. (2019) [83] | Bangladesh | 6 | Cross-sectional | Mental health program; peer-peer acupressure and mindful breathing techniques improved mental health, empowered self-care, and promoted caring relationships among Rohingya refugees | X | X | | X | |
| Tarannum et al. (2019) [50] | Bangladesh | 6 | Case study | Mental health program; mhGAP program was feasibly integrated into primary health care and led to 1,200 consultations in 9 months | X | X | | X | |
| Tol et al. (2020) [37] | Uganda | 2 | Cluster randomized control trial | Mental health program; guided-self help reduced psychological distress among 613 participants | X | X | | X | X |
| Maternal and child health (n = 14, 12%) | | | | | | | | | |
| Amsalu et al. (2020) [31] | Multi-site | 4 | Pre/post descriptive study | Newborn care; Helping Babies Survive training program is feasible, acceptable, and effective in improving health worker knowledge across 3 refugee camps. | X | X | | X | |
| Ayoya et al. (2013) [58] | Haiti | 6 | Case study | Newborn care; 193 baby tents cared for 180,499 mother-infant pairs to limit breastfeeding interruptions following the earthquake | X | X | | X | |
| Azad et al. (2019) [57] | Bangladesh | 6 | Cross-sectional | Newborn care; wet nurses enhance the IYCF-E protocol | X | | | | |

(*Continued*)

**Table 1.** (Continued)

| Citation | Country | Evidence level | Study type | Object of implementation and findings | Determinants | Implementation strategies | Theories, models, or frameworks | Scope | Clinical outcomes |
|---|---|---|---|---|---|---|---|---|---|
| Barua et al. (2022) | Bangladesh | 6 | Case study | Maternal health; 3,330 referrals made through a community-based transport intervention for emergency obstetric cases | X | X | | X | |
| Carrara et al. (2017) | Thailand | 3 | Quasi-experimental cohort implementation study | Newborn care; implementing micronutrient fortified flour for pregnant refugee women improved outcomes among 987 newborns | X | X | | X | X |
| Dozio et al. (2020) | Cameroon | 3 | Cohort implementation study | Newborn care; Baby-friendly spaces improve wellbeing of peripartum women in refugee camps through increased lactation and decreased stress | X | X | | X | X |
| Purdin et al. (2009) [86] | Pakistan | 4 | Cohort implementation study | Peripartum care; emergency obstetric centers improved maternal mortality, perinatal coverage, and number of skilled-attendant births | X | X | | X | X |
| Morris et al. (2012) [33] | Uganda | 2 | Randomized control trial | Newborn care; for 70 IDP mothers, combining psychosocial interventions with pre-existing feeding programs increased positive mother to newborn interactions and maternal mood | X | X | | X | X |
| Mullany et al. (2008) [84] | Myanmar | 6 | Case study | Maternal health; 300 community-level providers were successfully trained in maternal services | X | X | | X | |
| Rijken et al. (2009) [55] | Thailand | 6 | Qualitative cross-sectional | Prenatal care; local trained health workers obtained accurate fetal biometry measurements with OB ultrasound among 349 refugees at the Thai-Burmese border | X | X | | X | |
| Sami et al. (2017) [67] | South Sudan | 6 | Case study | Newborn care; training can increase health workers' knowledge on neonatal health | X | X | | X | |
| Sami et al. (2021) | South Sudan | 6 | Mixed methods pre/post descriptive study | Newborn care; newborn interventions are feasible via two-day simulation and supplies provision but do not lead to community adoption of interventions | X | X | | | |

(*Continued*)

**Table 1.** (Continued)

| Citation | Country | Evidence level | Study type | Object of implementation and findings | Determinants | Implementation strategies | Theories, models, or frameworks | Scope | Clinical outcomes |
|---|---|---|---|---|---|---|---|---|---|
| Sarker et al. (2020) [74] | Bangladesh | 6 | Qualitative | CFIR should be supplemented with context and security domains for use in humanitarian contexts | X | | X | | |
| Talley et al. (2013) | Haiti | 6 | Retrospective review | Newborn care; baby tents provided formula to 590 infants | X | X | | X | |
| Primary care / Non-communicable disease (n = 12, 10%) | | | | | | | | | |
| Alemu et al. (2022) | Ethiopia | 6 | Case study | Primary and mental care: Public Health Emergency Operation center provided health care to 33,410 IDPs including mental, antenatal, and cataract care | X | X | | X | |
| Ansbro et al. (2021) [52] | Jordan | 4 | Cohort implementation study | NCD treatment; primary-level NCD care for Syrian refugees found good clinical outcomes and high cost when evaluated with RE-AIM | X | X | X | X | X |
| Ansbro et al. (2023) | Lebanon | 4 | Prospective pe-post implementation study | NCD treatment; 418 Syrian refugees switched to fixed dose combination therapy for cardiovascular disease prevention | X | X | | X | X |
| Bile et al. (2011) [47] | Pakistan | 6 | Case study | Health services delivery; government and World Health Organization organized a health cluster of 46 humanitarian partners to reinforce the health system following natural disaster | X | X | | | |
| Bile et al. (2010) [48] | Pakistan | 6 | Case study | Primary and infectious care; Implementing the health cluster disease warning system controlled disease outbreaks, while health service package ensured delivery of primary and emergency health interventions | X | X | X | | |
| Ehiri et al. (2014) [41] | Multi-site | 5 | Systematic review | Primary care; training lay refugees as health workers increased service coverage, disease knowledge, and treatment protective behaviors | X | X | | X | |
| Kayali et al. (2019) | Lebanon | 4 | Retrospective cohort study | NCD treatment; Model for diabetes and hypertension care was feasible and led to controlled blood sugar and pressure | X | X | | X | X |

(*Continued*)

**Table 1.** (Continued)

| Citation | Country | Evidence level | Study type | Object of implementation and findings | Determinants | Implementation strategies | Theories, models, or frameworks | Scope | Clinical outcomes |
|---|---|---|---|---|---|---|---|---|---|
| Mahn et al. (2008) [49] | Myanmar | 6 | Case study | Health service delivery; health system building was successful for IDPs in Myanmar through cross border local-global partnerships | X | X | | | |
| Murphy et al. (2022) [30] | Lebanon | 6 | Qualitative (nested) | Novel medication therapy; fixed-dose combination therapy for atherosclerosis was considered acceptable and feasible among Syrian refugees | X | | | | |
| Sethi et al. (2017) [85] | Lebanon | 4 | Cohort implementation study | NCD treatment; 500 refugee volunteers cared for 387 patients with chronic disease | | X | | X | |
| Shortall et al. (2017) [65] | Greece | 6 | Retrospective review | Primary care; a ferry-based health clinic provided flexible and multidisciplinary care for refugees with continuous reassessment, mobilization, and adaptation | X | X | | X | |
| Sibai et al. (2020) [87] | Lebanon | 3 | Cohort implementation study | NCD screening; 1876 patients screened for NCDs with high satisfaction but barriers to scaling | X | X | X | X | |
| Health technology (n = 11, 9%) | | | | | | | | | |
| El-Halabi et al. (2023) | Jordan | 3 | Non-randomized controlled trial | mHealth; vaccine app among 471 Syrian refugees slightly increased vaccine follow-up | X | X | | X | |
| Khader et al. (2022) [62] | Jordan | 6 | Qualitative cross-sectional | mHealth; Children Immunization App disseminated health information to 1100 children in refugee camps | X | X | | X | |
| Klabbers et al. (2023) | Uganda | 6 | Case study | Phone calls; Over 10,000 public health messages were disseminated to refugees on COVID-19 symptoms and exposures | X | X | | X | |
| Lyles et al. (2023) | Kenya | 6 | Case study | mHealth; adapted app reported 7,918 consultations with high satisfaction for NCD management | X | X | | X | |
| McEwen et al. (2024) | Lebanon | 6 | Case study | Phone-based therapy; t-CETA was feasible, but both created and solved logistical challenges | X | X | | | |

(*Continued*)

**Table 1.** (Continued)

| Citation | Country | Evidence level | Study type | Object of implementation and findings | Determinants | Implementation strategies | Theories, models, or frameworks | Scope | Clinical outcomes |
|---|---|---|---|---|---|---|---|---|---|
| Mitchell-Gillespie et al. (2020) [61] | Jordan | 6 | Cohort implementation study | Physical and occupational health; Telehealth successfully supported rehabilitation workers | X | X | X | | |
| O'Laughlin et al. (2021) [63] | Uganda | 4 | Cohort implementation study | mHealth; Phone call or text to encourage service linkage was effective for follow-up HIV care | X | | | X | |
| Rossi et al. (2009) [60] | Lebanon | 6 | Case study | Hospital IT; improved tracking through case-mix hospital information system for Palestinian refugees | X | X | | | |
| Sampson et al. (2023) | Nigeria | 4 | Case study | mHealth; mobile technology intervention increased family planning, spousal consent, and contraceptive prevalence among 103 IDPs | X | X | | X | X |
| Shaikh et al. (2008) [64] | Somalia | 6 | Case study | Information mapping; use of global information systems facilitated 3,095 mobile clinic visits for IDPs in Somalia | X | X | | X | X |
| Wilton et al. (2023) | Multi-site | 6 | Case study | mHealth; Reach Up and Learn with interactive voice responses improved responsiveness to children | X | X | | X | |
| Sexual and reproductive health (n = 7, 6%) | | | | | | | | | |
| Beeman et al. (2023) | Uganda | 6 | Case study | Menstrual health; Cocoon Mini showed high acceptability among 175 refugees to facilitate secure space for menstrual health | X | X | | X | |
| Curry et al. (2015) [79] | Multi-site | 6 | Case study | Family planning; SAFPAC was implemented with varying success across five refugee settings | X | X | | X | |
| Curry et al. (2015) [80] | Multi-site | 6 | Case study | SAFPAC reached 52,616 new users of modern contraceptives | X | X | | X | X |
| Fetters et al. (2020) [69] | Bangladesh | 6 | Case study | Family planning and sexual health services; 300 health workers from 37 facilities were trained to care for 31,000 patients | X | X | | X | |
| O'Connell et al. (2022) | Ethiopia | 6 | Case study | Sexual health services; implementation among 4,500 IDPs demonstrated challenges and success metrics | X | X | | X | |
| Von Roenne et al. (2010) | Guinea | 6 | Case study | Sexual health services; reproductive services can be implemented by peer refugees with sustained funding and assistance | X | X | | X | |

(*Continued*)

**Table 1.** (Continued)

| Citation | Country | Evidence level | Study type | Object of implementation and findings | Determinants | Implementation strategies | Theories, models, or frameworks | Scope | Clinical outcomes |
|---|---|---|---|---|---|---|---|---|---|
| Warren et al. (2015) [43] | Multi-site | 5 | Systematic review | Sexual health services; several interventions were effective for young people's sexual health in humanitarian and LMIC settings | X | | | X | |
| Ophthalmology (n = 2, 2%) | | | | | | | | | |
| Ahmed et al. (2020) [53] | Bangladesh | 6 | Cross-sectional cohort study | Successful vision screening campaign determined that eye disease was higher among Rohingya refugees than host population | X | X | | X | |
| Vincent et al. (2006) [54] | Thai-Burma border | 6 | Cross-sectional cohort study | Health worker training facilitated sustainable, low-cost spectacle provision during 7,219 refugee visits within a challenging environment and geography | X | X | | X | |
| Surgery (n = 2, 2%) | | | | | | | | | |
| Moreau et al. (2020) [59] | Jordan | 6 | Case study | Reconstructive surgery; 3D technology and telemedicine provided facial orthoses for 24 patients and limb prostheses for 29. | X | X | | X | |
| Sechriest et al. (2008) [66] | Southeast Asia | 6 | Case study | Orthopedic surgery; Navy and Project HOPE cared for complex orthopedic patients on the US Naval Ship Mercy and complemented local medical teams | X | X | | X | |
| Pharmacy (n = 1, 1%) | | | | | | | | | |
| Del Cacho et al. (2021) | Algeria | 6 | Case study | Successful implementation of a pharmacy distribution and training program | X | | | X | |
| Nutrition (n = 1, 1%) | | | | | | | | | |
| Rah et al. (2012) [42] | Multi-site | 4 | Systematic review | Nutrition program; self-reported acceptability and adherence to use of micronutrient powders was acceptable but varied across sites | X | X | | X | X |
| | | | | Totals | 110 (95%) | 106 (91%) | 10 (9%) | 99 (85%) | 37 (32%) |

## Objects of implementation

Implemented programs reflected topic areas (Table 1). The most frequently implemented interventions were mental health therapy programs (25%) and vaccination services (16%). Mahmuda et al. described a group integrative adaptation therapy among Rohingya refugees [45] and Sheikh et al. evaluated oral polio distribution within Kenyan refugee camps [46].

Almost all programs focused on a single area with a vertical care delivery approach, as only 7 studies reported on enhancing health services delivery in a horizontal approach that prioritized health systems strengthening [41, 47–51]. Examples of other objects of implementation included novel medication administration programs [52], eye disease screening [53, 54], obstetric ultrasound training [55], substance use screening [56], and breastfeeding or nutrition programs [57, 58]. Seven studies utilized novel technologies in their programs. Moreau et al. from MSF in Jordan used 3D technologies and telemedicine to customize and provide facial ortheses and limb prostheses to 53 patients [59]. Others established technology infrastructure in hospitals for health data [60, 61], mHealth screenings and reminders through text-messaging or applications [62, 63], and global information systems for disease control [64]. Two studies described floating boat hospitals which delivered preventive care in Greece [65] and orthopedic surgical services in Southeast Asia [66].

## Determinants

Almost all studies (95%) reported key determinants–barriers and facilitators–to program implementation. In total, 38 unique barriers were identified among the 116 studies (Table 2). Resource limitation was the most frequently mentioned barrier (30%). This included a lack of diagnostic tools, treatment therapy, or health access infrastructure such as cell phones for mobile health programs. Second was a shortage of trained health workers (24%), due to limited training, high staff turnover, and poor retention working in humanitarian settings. High staff turnover posed a barrier to implementing newborn health interventions in South Sudan [67]. Security risks from conflict (24%), environmental conditions including natural disasters and extreme weather (21%), and negative perceptions of the migrant community towards the intervention (17%) were all commonly reported. Some barriers were likely unique to migrant populations in humanitarian settings. These included nonexistent health insurance coverage (1%), inability to access public health systems (1%), and the transitory nature of migrant populations, which limited patient follow up, increased dropout, and may have demotivated transitory migrants from accessing services at all (16%). In Haiti, the evaluation of rapid monitoring vaccination campaigns was limited by continual migration out of and between multiple IDP camps [68].

Common facilitators to successful program implementation were also described. Overall, 29 facilitating factors were identified (Table 2). The most common were health worker availability (25%), referring to physical presence and appropriate training to implement specific health programs by these individuals, and the effective leveraging of strong pre-existing partnerships (25%), which included collaboration with local NGOs, religious and community leaders, or health facilities. Fetters et al. demonstrated this strategy to train 300 health workers in sexual and reproductive health care in Bangladesh refugee camps [69]. Additional external facilitators included positive perceptions of the intervention from the migrant population (20%), availability of a pre-established health system to build on (19%), and alignment with political priorities of a supervisory organization, particularly the government (17%). Internal facilitators specific to interventions were also identified, including being low-cost with minimal resource requirements and culturally appropriate. For example, programs which reported success adapted interventions to local language and customs in consultation with community experts. In Uganda, Jordan, and Iraq, mental health programs were adapted to be self- or peer-administered to facilitate their implementation despite a lack of trained mental health professionals [70–72]. The fidelity of interventions was occasionally adapted successfully to low-resource contexts (12%), such as using vaccines with high thermostability to mitigate freezer supply chain barriers [73] or combining novel drug therapies to reduce distribution frequency

**Table 2. Determinants including facilitators and barriers to implementing migrant health services in humanitarian settings.**

| Facilitator | Count | Frequency | Barrier | Count | Frequency |
|---|---|---|---|---|---|
| Health worker availability | 29 | 25% | Resource limitations | 35 | 30% |
| Pre-existing partnerships | 29 | 25% | Limited trained health workers | 28 | 24% |
| Positive perceptions towards intervention | 23 | 20% | Security risks | 28 | 24% |
| Pre-established health system | 22 | 19% | Environmental conditions | 24 | 21% |
| Alignment with political priorities | 20 | 17% | Negative perceptions towards the intervention | 20 | 17% |
| High-income institution or other international partnership | 16 | 14% | Migratory population | 18 | 16% |
| Culturally appropriate intervention | 14 | 12% | Low knowledge or health literacy | 18 | 16% |
| Adaptability of intervention | 14 | 12% | Low data quality | 16 | 14% |
| Clear and standard procedures / strong evidence base | 12 | 10% | Inadequate infrastructure including physical space closure | 16 | 14% |
| Low-cost / free to population | 8 | 7% | Time constraints | 15 | 13% |
| Trust in health workers | 6 | 5% | Cost | 13 | 11% |
| Established resource supply chains | 6 | 5% | Lack of cultural appropriateness | 11 | 9% |
| Familial support | 5 | 4% | Language barriers | 10 | 9% |
| Minimal training required for intervention | 5 | 4% | Geographic barriers | 10 | 9% |
| Flexible and sufficient funding | 4 | 3% | Intervention bulkiness or complexity | 9 | 8% |
| Expert consultants | 4 | 3% | Funding restrictions | 9 | 8% |
| Intervention effectiveness | 4 | 3% | Technological failures or lack of network connectivity | 9 | 8% |
| Minimal resource requirements | 3 | 3% | Lack of evidence or unspecified clinical guidelines | 7 | 6% |
| Political environment facilitating implementation | 3 | 3% | COVID-19 travel restrictions/lockdowns | 7 | 6% |
| Communication systems | 3 | 3% | Sex disparities | 7 | 6% |
| Ease of access; 24/7 service availability | 3 | 3% | Ethical dilemmas for humanitarian research including randomization and informed consent | 6 | 5% |
| Intervention scalability | 2 | 2% | Competing political priorities, policy resistance | 5 | 4% |
| Site similarity | 2 | 2% | Population diversity | 5 | 4% |
| Availability of infrastructure | 2 | 2% | Overly complex health systems | 4 | 3% |
| Additional incentives from intervention | 2 | 2% | Mistrust towards health system or workers | 4 | 3% |
| Social interconnectedness | 2 | 2% | Stigma | 4 | 3% |
| Good baseline population health | 1 | 1% | Demotivated or disinterested participants or staff | 4 | 3% |
| Stable conditions with minimal security risks | 1 | 1% | Limited central leadership | 3 | 3% |
| High health literacy | 1 | 1% | Duplicated services | 3 | 3% |
| | | | Poverty | 3 | 3% |
| | | | Lack of cooperativity or trust among organization leadership | 2 | 2% |
| | | | Narrow intervention scope | 2 | 2% |
| | | | Choosing between health and work | 1 | 1% |
| | | | Unavailability of decision makers | 1 | 1% |
| | | | Limited insurance coverage | 1 | 1% |
| | | | Need to de-implement incorrect practices | 1 | 1% |
| | | | Lack of subject area expertise | 1 | 1% |
| | | | Non-citizen status | 1 | 1% |

[30]. Many facilitators and barriers mirrored one another across studies, including throughout diverse regions and topic areas. These juxtaposed factors included positive vs. negative migrant and other stakeholder perceptions of interventions, high vs. low population health literacy, affordable vs. expensive interventions, and trust vs. mistrust in the health system. Three studies

undertook a determinant analysis with the primary objective of identifying barriers and facilitators rather than strategies or outcomes of program implementation [57, 74, 75].

## Implementation strategies

To fulfill our primary review objective, we extracted and categorized strategies used to implement health services among displaced populations in humanitarian settings. In total, 35 unique strategies were identified among 106 (91%) studies that made any mention of implementation strategy (Table 3). Bundles of two or more implementation strategies were used in 102 studies (88%). The most common strategy was capacity building, which overwhelmingly included health worker training (44%). For example, capacity building through community and health

**Table 3. Implementation science strategies used by humanitarian actors to implement and scale health services among migrants in humanitarian settings.**

| Implementation strategy | Count | Frequency |
|---|---|---|
| Capacity building | 51 | 44% |
| Stakeholder engagement | 41 | 35% |
| Information dissemination and community education | 38 | 33% |
| Feedback and evaluation mechanisms | 33 | 28% |
| Collaborating with local partners | 32 | 28% |
| Needs assessment | 26 | 22% |
| Local champions | 26 | 22% |
| Task-shifting to lower-level providers such as community health or lay providers | 23 | 20% |
| Integration into existing health systems | 22 | 19% |
| Intervention adaptation, including local contexts and language or resource constraints | 21 | 18% |
| Supervision | 20 | 17% |
| Integrating mobile delivery components | 14 | 12% |
| Establishing supply chains and resources acquisition methods | 12 | 10% |
| Establishing infrastructure or designating specialized physical space | 12 | 10% |
| Interdisciplinary approach | 10 | 9% |
| Integrating technology | 10 | 9% |
| Train-the-trainer dissemination models | 10 | 9% |
| Continual or iterative support | 10 | 9% |
| Centralized coordination | 9 | 8% |
| Identifying and targeting high-risk populations | 8 | 7% |
| Promoting local autonomy and ownership / intervention co-design | 8 | 7% |
| Piloting | 7 | 6% |
| Accurate data monitoring, including through usage of health records | 6 | 5% |
| Raising political support | 5 | 4% |
| Facilitating intervention implementation through group activity | 4 | 3% |
| Game play | 4 | 3% |
| Partnership building with referral sites | 3 | 3% |
| Prioritizing low-cost materials | 2 | 2% |
| Evidence gathering | 2 | 2% |
| Bundling multiple evidence-based interventions simultaneously | 2 | 2% |
| Leveraging social mobilization for engagement | 2 | 2% |
| Remuneration | 2 | 2% |
| Motivating health workers | 1 | 1% |
| Protecting consent through voluntary participation | 1 | 1% |
| Intentional interprofessional recruitment | 1 | 1% |

worker training in Tanzania allowed for integration of a prevention of mother to child HIV transmission program into existing health systems [76]. Stakeholder engagement was the second most common strategy (35%) and included coordination and collaboration among diverse groups including government agencies [42], community and religious leaders [77], migrant patients [78], and product developers. Information dissemination and community education was the third most common (33%), followed by feedback and evaluation mechanisms (25%). The most common method used for feedback was checklists, which were used to facilitate service delivery for contraception [79, 80], diphtheria toxin [81], and meningitis and polio vaccines across eight countries [46, 82].

Many strategies highlighted the importance of local collaboration and human resource management, including collaborating with local partners (28%), local champions (22%), task-shifting to lower-level providers (20%), train-the-trainer models (9%), and community empowerment through intervention co-design (7%). In Bangladesh, Sullivan et al. used local collaboration, task-shifting, and a train-the-trainer distribution model to improve mental health perceptions of Rohingya refugees through acupressure and breathing techniques [83]. Intervention adaptability was an important theme which emerged, as adaptation often became necessary given changing contexts in resource constrained environments (18%). Specific actions included translation to local languages [38, 47, 58], mobile delivery components for geographically constrained and transitory populations [32, 53, 84, 85], and increasing cultural acceptability by accounting for sex differences [86]. Strategies less frequently utilized through these studies were remuneration (2%), social mobilization (2%), and motivating health workers (1%).

## Implementation science theories, models, and frameworks

Although most studies sought to identify determinants and implementation strategies, only 10 (8%) employed formal implementation science theories, models, or frameworks (Table 4). The most common framework employed was RE-AIM (Reach, Effectiveness, Adoption, Implementation, and Maintenance). This framework was used in 3 studies for occupational health and non-communicable disease in Jordan [52, 61] and as an analytic tool as part of an 11-study scoping review [40]. The Dynamic Sustainability Framework (DSF) [52], Consolidated Framework for Implementation Research (CFIR) [74], and Precede-Proceed [87] were all used once. Intervention mapping (IM) guided the design of two programs. Ravicz et al. used intervention mapping to create a mental health intervention among Rohingya refugees, while Borja et al. used IM to determine where to implement HIV linkage interventions within a complex Ugandan care system [51, 88]. Additional models included Design, Implementation, Monitoring, and Evaluation (DIME) [89], and frameworks specific to global and humanitarian health: the Global Vaccine Action Plan and Humanitarian Health Response Coordination Framework [48, 90]. While these models are less common in implementation science, they contained similar components to well-known frameworks, including establishing priorities, implementation plans, and evaluation metrics. Only one study measured implementation factors, including uptake, feasibility, and acceptability, as a primary outcome [89]. None of the studies compared implementation frameworks, but eight explicitly described how using a framework positively enhanced their implementation planning, process, or outcomes measurement.

No studies could be considered rigorously evaluated implementation optimization trials which used methods such as Sequential Multiple Assignment Randomized Trial (SMART) or Multiphase Optimization Strategy (MOST). None compared outcomes of different implementation strategies, as could be done with effectiveness-implementation hybrid designs. A complete list of study references is available in S2 Appendix.

**Table 4. Comprehensive information on the use of implementation science theories, models, and frameworks for implementing health services among migrants in humanitarian settings.**

| Study title | Theories, models, or frameworks | Justification for selection | Evaluation and key points of use |
|---|---|---|---|
| MSF experiences of providing multidisciplinary primary level NCD care for Syrian refugees and the host population in Jordan: an implementation study guided by the RE-AIM framework [52] | Reach, Effectiveness, Adoption, Implementation, Maintenance (RE-AIM) | • Used previously in LMICs<br>• Designed to facilitate translation of research into practice and improve reporting of key elements for implementation at individual and organizational levels<br>• Can be adapted to context<br>• Analyzed a range of outcomes including quality of care, cost-effectiveness, and unintended consequences within "effectiveness" domain<br>• Facilitated integrated mixed methods and a qualitative coding framework | RE-AIM proved a valuable tool to evaluate a complex intervention in a protracted humanitarian crisis setting. It allowed presentation of clinical outcomes, quality of life indicators, perceived benefits, unintended consequences, cost and economic outcomes, usability, and sustainability from multiple stakeholder analyses at the individual and organizational level in one coherent framework. Modifications to RE-AIM allowed users to collect more qualitative and cost-effectiveness data |
| Task-shifting for refugee mental health and psychosocial support: A scoping review of services in humanitarian settings through the lens of RE-AIM [40] | RE-AIM | • Designed to understand complex interventions<br>• Critical to guide program implementation in low-resource settings where infrastructure is limited for complex projects<br>• No studies included in the review used an implementation science framework | Employing RE-AIM allowed for a stronger analysis of the development, adaptation, and implementation of interventions than each single study independently. The application as a scoping review analysis was feasible and accomplished study aims. However, RE-AIM falls short of comprehensively describing complexities unique to humanitarian contexts and supporting health staff. |
| Sustainable support solutions for community-based rehabilitation workers in refugee camps: piloting telehealth acceptability and implementation [61] | RE-AIM and Dynamic Sustainability Framework (DSF) | • RE-AIM guided recruitment, training, system use, and data collection design<br>• Designed data collection to account for sustainability and scalability from the beginning<br>• First four components (RE-AI) prepared implementation for maintenance/sustainability phase, upon which DSF was initiated<br>• DSF facilitated improved fit and sustainability through iterative evaluation at intervention, practice setting, and ecological system levels | Represented a novel application of two frameworks to assess telehealth in a humanitarian setting. RE-AIM facilitated sustainability planning beginning with intervention design, measured participant satisfaction, and structured process evaluation. DSF maximized intervention optimization and fit. Both prioritized intervention scaling by soliciting participant insights on determinants for sustainability. The authors created an effective visual to understand applying RE-AIM and DSF at multiple system levels. |
| Child-centred, cross-sectoral mental health and psychosocial support interventions in the Rohingya response: a field report by Save the Children [88] | Intervention mapping (Save the Children Mental Health and Psychosocial Support Programming) | • Allowed for mapping of ecology at child, family, community, and society levels, and placed children at the center of programming<br>• Accounted for gender and power dynamics<br>• Incorporated cross-cutting thematic areas including health, nutrition, education, and security<br>• Specified intended implementation outcomes at caregiver and community levels<br>• Mapped potential interventions to specific levels, topic areas, and desired outcomes with an accessible visual<br>• Facilitated clear guidelines of what was within the organization's scope and when to refer to other services | Intervention mapping (IM) facilitated organizing 20 interventions at specific levels in a complex system with multiple actors and gender and power dynamics. The planning process centered children as the primary feedback source and led to excellent data on intervention determinants, particularly evidence gaps and implementation barriers. IM guided where to apply interventions but fell short of how to select the best evidence-based practices or evaluate interventions once implemented. |

*(Continued)*

**Table 4.** (Continued)

| Study title | Theories, models, or frameworks | Justification for selection | Evaluation and key points of use |
|---|---|---|---|
| Using Intervention Mapping methodology to design an HIV linkage intervention in a refugee settlement in rural Uganda [51] | Intervention mapping / PRECEDE | • Six step method to develop evidence based interventions by engaging diverse stakeholders with an iterative step-by-step process<br>• Key focus was stakeholder and community ownership and engagement<br>• Used previously in LMICs for programs on sexually transmitted infections and treatment adherence<br>• A systematic review demonstrated that IM-guided programs had improved uptake versus controls<br>• Built on previous models to bridge theory, evidence, and practical implementation for health promotion programs<br>• Accounted for diverse socioecological factors affecting implementation including environmental and behavioral determinants and factors<br>• PRECEDE guided the needs assessment and applied theories to health interventions | IM provided an inclusive, efficient method for integrating community members and program implementers in intervention planning in a humanitarian setting. It defined the program's scope, outcomes, design, and production with sub-analysis for attitudes and values, skills and self-efficacy, knowledge, and outcome expectations, change methods, and practical applications. It successfully developed a program to integrate services for HIV and noncommunicable diseases at community sites. However, the use of IM in this study did not include implementing the program or evaluating outcomes. |
| Effective maternal, newborn and child health programming among Rohingya refugees in Cox's Bazar, Bangladesh: Implementation challenges and potential solutions [74] | Consolidated Framework for Implementation Research (CFIR) | • Allowed for categorizing implementation challenges and solutions according to its domains: implementation process, inner setting, and outer setting<br>• Sufficiently flexible that authors adapted CFIR to incorporate two new domains: context and security<br>• Differentiated challenges and solutions at multiple levels (inner vs. outer setting) | CFIR was used to guide thematic analysis of qualitative interviews on challenges and potential solutions for implementation of maternal, newborn, and child health. The framework was considered an asset for designing implementation but did not implement the program. The researchers' supplementing of CFIR with additional domains for context-specific needs was novel and could aid other humanitarian settings. |
| Lessons learned in the provision NCD primary care to Syrian refugee and host communities in Lebanon: the need to 'act locally and think globally" [87] | Precede-Proceed | • Used for monitoring and evaluation<br>• Provided a holistic view for multifaceted outcomes and multicomponent interventions with process, behavioral, and environmental targets<br>• Accounted for predisposing, reinforcing, and enabling factors which linked to behavioral and environmental targets | Overall, the Precede-Proceed model was not explicitly evaluated or commented upon by study authors. It was used to decide intervention targets and for evaluation of an already implemented program. Authors provided a helpful visual of their use of Precede-Proceed. |
| An evaluation of a common elements treatment approach for youth in Somali refugee camps [89] | Design, Implementation, Maintenance, Evaluation (DIME) | • Designed to identify local mental health problems qualitatively, guide selection, adaptation, and testing of instruments and interventions, and evaluate provided services with local providers<br>• Demonstrated high cultural competency by incorporating local idioms of symptoms to validate data instruments<br>• Evaluation of acceptability, applicability/fit, feasibility, treatment facilitators, and barriers from diverse perspectives including implementers, users, and partners | DIME guided the design, implementation, and assessment of a mental health program with high cultural competency using comprehensive mixed methods. The framework's flexibility allowed for methodologic adaptations including a "common elements" treatment approach to intervention implementation with varied dosing, elements, and order. DIME quantified intervention uptake and identified facilitators, barriers, and solutions to implementation factors. DIME's lack of rigidity offered less structure to guide intervention design and evaluate implementation compared with other models. |

(*Continued*)

**Table 4.** (Continued)

| Study title | Theories, models, or frameworks | Justification for selection | Evaluation and key points of use |
|---|---|---|---|
| Integrating immunisation services into nutrition sites to improve immunization status of internally displaced persons' children living in Bentiu protection of civilian state, South Sudan [90] | Global Vaccine Action Plan | • Endorsed by UNICEF and the World Health Organization<br>• Aligned with study topic area and objectives<br>• Strong evidence base for effectiveness of integrating vaccines into other health services | The Global Vaccine Action Plan guided key decisions for how to design the intervention, particularly integration with preexisting health services. The authors do not describe how the other six points of the GVAP were used or if there were successes in implementation from using the framework. |
| Learning through crisis: development and implementation of a health cluster strategy for internally displaced persons [48] | Humanitarian Health Response Coordination Framework | • Strong evidence for success in other contexts<br>• Addressed a priority area for response and scaling as identified by WHO<br>• Considered favorable and acceptable by all key actors<br>• Standardized operational guidelines across diverse actors and service providers<br>• Designed to provide clearer avenues for effective coordination, joint planning, distribution of roles and responsibilities, resource mobilization, the creation of operational synergies, complementarities, and shared accountability between partners | A multidisciplinary, 13 health response intervention framework, the Humanitarian health response coordination framework coordinated roles and responsibilities of large organizations in a complex emergency. It organized 46 national and international organizations to generate effective and cohesive partnerships, which coordinated a rapid needs assessment, disease early warning system and mitigation program, essential services package, mass vaccinations, maternal and child health, nutrition surveillance, health education, medicinal supply management, WASH infrastructure, psychosocial support, and assistance for those with disabilities. The strategy was an actionable tool, able to resolve key challenges and create cooperation, effective communication, collective efforts for health system strengthening, and shared compliance with organizational and logistical standards. |

## Discussion

In this scoping review, we found that common implementation science strategies were often utilized to deliver health care to migrants in humanitarian settings. Though studies came from diverse regions, populations, and topic areas, determinants (barriers and facilitators) and implementation strategies were well described, shared, and could be defined and categorized among the studies. The use of formal implementation models, frameworks, or theories to guide implementation and evaluation of health service programs was exceptionally rare. Models that were used included RE-AIM, Intervention Mapping, CFIR, and Precede-Proceed. The use of these models was associated with improved intervention planning, implementation, and evaluation. This main finding suggests that implementation science is an underutilized approach and promising option for humanitarian actors seeking to design, implement, and evaluate health programs in a more systematic, rigorous, and universally understood way.

We found that objects of implementation–the services being delivered–generally reflected the major health needs of migrants: mental health, infectious disease, and maternal/child health. In general, services focused on prevention including vaccine campaigns, mental health screening and therapy, and prenatal care. While a few studies were treatment oriented, particularly for HIV and mental health disorders, there was little cost-effectiveness or clinical evaluation metrics to influence policy and funding decisions. Additionally, our results showed a clear gap in subspeciality and procedural care including surgery, ophthalmology, physical and occupational therapy, and nutrition. There is now robust data demonstrating the health needs of migrants in each of these areas [91, 92]. This suggests the need for more efforts to understand

effective implementation of these services, which often require increased coordination, resources, and patient monitoring and follow-up. Study rigor was quite low, with more than half being cross-sectional case studies and less than one-third reporting clinical outcomes. An additional gap was found with geographic representation of studies, with most coming from Africa, the Middle East, and Southeast Asia, and very few from the Americas. More work may be needed to understand context-specific implementation for Western migrant contexts including with asylum seekers at the US-Mexico border, Venezuelan refugees, and those fleeing violence throughout Central America and Haiti [93].

Nearly every study identified key determinants–barriers or facilitators–to implementing health services. Although we included studies from 37 countries and 11 topic areas, the barriers and facilitators were universal enough to be grouped into 38 and 29 unique items, respectively. The shared experiences of determinants across diverse settings is notable. These findings may help humanitarians planning projects to better identify, define, and categorize determinants they encounter in project implementation. Determinants included external (security risks, environmental conditions, political priorities) and internal factors (organization funding, intervention adaptability, cost and resource requirements), many of which career humanitarians have anecdotally described [94]. Though many of these determinants are likely common in low-resource environments, some were specific to migrant populations, including a lack of follow up from continued migration, demotivation for participants to engage while being in a transitory status, security risks, and ethical dilemmas related to study methods including randomization. While most studies identified determinants, few prospectively accounted for them in designing their implementation approach; rather, they were retrospectively described as they appeared to the study teams throughout service delivery. This retrospective approach, while less effective, seems to be common across humanitarian and conflict settings [17]. By using validated implementation science models from the beginning, humanitarian actors may be more likely to identify and account for determinants prior to project implementation, allowing them to mitigate barriers and enhance facilitators for greater project success.

Implementation strategies were frequently reported. More than 90% of studies mentioned at least one strategy, and nearly all of these utilized multiple strategies. Similar to determinants, strategies employed were common despite diverse geographic and health specialties across studies. The most described strategies addressed human capacity building: health worker training and availability, stakeholder engagement, collaborating with local partners, and empowering local champions. Other strategies were process oriented: conducting needs assessments, information dissemination, feedback and evaluation mechanisms, and establishing specific infrastructure and supply chains. Strategies emphasized integration of the local community into planning and programming. Beyond stakeholder engagement, studies emphasized local autonomy and collaboration, co-designing interventions and implementation plans, adapting interventions and approaches to local and culturally appropriate methods, and raising meso- and macro-level political support for initiatives. These results suggest moving from the traditional approaches of aid as service delivery to instead empowering community members and leaders in skills and resource capital building while in humanitarian contexts [95, 96]. This highlights an opportunity to apply tools designed for stakeholder engagement when co-designing implementation, such as the 7P's framework [97]. Strategies dealing with cost were described, but less commonly than human capacity building or adaptability. These approaches included remuneration and using low-cost materials, but only encompassed three studies.

Despite ubiquitous descriptions of determinants and implementation strategies, formal implementation theories, models, or frameworks were rarely employed. RE-AIM, Intervention

mapping, CFIR, Precede-Proceed, DSF, and global-humanitarian specific frameworks were used for only 8% of studies. While leveraging implementation models for humanitarian aid could be highly beneficial, it is unsurprising that this approach is not more widely practiced. First, implementation science as a field itself is still maturing, particularly into areas outside of well-resourced health systems [98]. Implementation science in global health, for example, is rapidly growing due to its unique ability to address specific issues that have long affected global health and could provide helpful examples for those working in humanitarian aid [14]. Second, research among humanitarian aid organizations is still nascent. While there have been calls for increasing scholarship to optimize humanitarian aid, most organizations lack the resources and expertise to conduct independent evaluations [99]. Humanitarian aid-academic medical research partnerships have been proposed and could also help ameliorate this gap [99]. Additionally, smaller groups could replicate methods proven to be feasible from organizations known for research productivity including Médecins Sans Frontières and the World Health Organization [100]. Notwithstanding the novelty of the use of implementation science in designing and evaluating humanitarian aid interventions, there are efforts to promote this scholarship [101]. Recently, a rapid scoping review described implementation science strategies among resettled refugees, demonstrating barriers, strategies, and evaluation methods for refugee resettlement [16]. Barriers included time and funding constraints, workflow disruptions, and language barriers. Strategies were adapting interventions to local context, training stakeholders, and iterative evaluation. Our results complement this study by presenting results for migrants still in transit. While many of the common strategies are shared between the studies, ours revealed unique results specific to implementation in a humanitarian context, including mobile delivery components, integration into pre-existing health systems, and collaborating with local partners. In summary, determinants and health implementation strategies among transitory migrants may be quite distinct from those being resettled.

## Limitations

This scoping review had several limitations. First, we limited our search to published, peer-reviewed articles within the last 20 years. While this was done intentionally to focus on rigorous studies with novel health services, excluding gray literature or reports from humanitarian organizations could limit results. Second, this study was limited in its scope to focus on migrants in humanitarian settings. Studies which reported on other types of global health programs, programs that may be similar but distinct from humanitarian contexts, were intentionally excluded. Contextualizing our results within this literature could help to enhance health services for other vulnerable populations and common challenges. Third, this review used scoping methods and was not a systematic review or meta-analysis. Therefore, we were unable to make claims about clinical outcomes or effectiveness across various health delivery strategies.

## Conclusions

In this scoping review of migrants in humanitarian settings among diverse geographic regions and health topic areas, we found common and shared implementation determinants and strategies, varied evidence rigor and outcomes, and that use of formal implementation models and frameworks was rare. Implementation science provides an exciting opportunity for more effectively planning, implementing, and evaluating health services programming for migrants in humanitarian contexts.

## Supporting information

**S1 Appendix. Comprehensive data extraction from scoping review studies for displaced migrants in humanitarian settings.** This table presents all the extracted data from studies included in the review including study rigor, descriptors, determinants, strategies, location, and use of implementation science theories, models, or frameworks for each study. (DOCX)

**S2 Appendix. References for studies included in the scoping review presented in order as they appear in Table 1.** This file provides a complete list of studies included in the scoping review, in order of appearance of Table 1.
(DOCX)

## Author Contributions

**Conceptualization:** Christopher W. Reynolds.

**Data curation:** Christopher W. Reynolds, Jennifer Y. Rha, Allison M. Lenselink, Dhanya Asokumar, Laura Zebib, Gurpreet K. Rana, Francesca L. Giacona, Nowshin N. Islam, Sanjana Kannikeswaran, Kara Manuel, Allison W. Cheung, Maedeh Marzoughi, Michele Heisler.

**Formal analysis:** Christopher W. Reynolds, Jennifer Y. Rha, Allison M. Lenselink, Dhanya Asokumar, Laura Zebib, Gurpreet K. Rana, Francesca L. Giacona, Nowshin N. Islam, Sanjana Kannikeswaran, Kara Manuel, Allison W. Cheung, Maedeh Marzoughi, Michele Heisler.

**Investigation:** Christopher W. Reynolds, Gurpreet K. Rana, Michele Heisler.

**Methodology:** Christopher W. Reynolds, Gurpreet K. Rana, Michele Heisler.

**Project administration:** Christopher W. Reynolds, Gurpreet K. Rana, Michele Heisler.

**Resources:** Gurpreet K. Rana, Michele Heisler.

**Software:** Gurpreet K. Rana.

**Supervision:** Christopher W. Reynolds, Gurpreet K. Rana.

**Validation:** Christopher W. Reynolds.

**Visualization:** Gurpreet K. Rana.

**Writing – original draft:** Christopher W. Reynolds.

**Writing – review & editing:** Jennifer Y. Rha, Allison M. Lenselink, Dhanya Asokumar, Laura Zebib, Gurpreet K. Rana, Francesca L. Giacona, Nowshin N. Islam, Sanjana Kannikeswaran, Kara Manuel, Allison W. Cheung, Maedeh Marzoughi, Michele Heisler.

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
