## [Decision Letter · Decision Letter 0]

26 Jun 2024

PGPH-D-24-00863

Innovative strategies and implementation science approaches for health delivery among migrants in humanitarian settings: a scoping review

Dear Dr. Reynolds,

Thank you for submitting your manuscript to PLOS Global Public Health. After careful consideration, we feel that it has merit but does not fully meet PLOS Global Public Health’s publication criteria as it currently stands. Therefore, we invite you to submit a revised version of the manuscript that addresses the points raised during the review process.

We look forward to receiving your revised manuscript.

Kind regards,

Nnodimele Onuigbo Atulomah, PhD

Academic Editor

Journal Requirements:

Additional Editor Comments (if provided):

The review has been very encouraging only minor corrections are required for this manuscript. Please note the recommendations for revision by the third reviewer.

Reviewers' comments:

Reviewer's Responses to Questions

**Comments to the Author**

1. Does this manuscript meet PLOS Global Public Health’s publication criteria? Is the manuscript technically sound, and do the data support the conclusions? The manuscript must describe methodologically and ethically rigorous research with conclusions that are appropriately drawn based on the data presented.

Reviewer #1: Yes

Reviewer #2: Yes

Reviewer #3: Yes

2. Has the statistical analysis been performed appropriately and rigorously?

Reviewer #1: Yes

Reviewer #2: Yes

Reviewer #3: N/A

3. Have the authors made all data underlying the findings in their manuscript fully available (please refer to the Data Availability Statement at the start of the manuscript PDF file)?

Reviewer #1: Yes

Reviewer #2: Yes

Reviewer #3: Yes

4. Is the manuscript presented in an intelligible fashion and written in standard English?

Reviewer #1: Yes

Reviewer #2: Yes

Reviewer #3: Yes

5. Review Comments to the Author

Reviewer #1: The article is timely, well researched and excellently written. The objectives, methods and findings are well stated and justified. It is applicable to ongoing humanitarian efforts in different parts of the world, especially in resource-prive countries which strikingly are the worst hit by emergencies that lead to forced migration of their populations. I agree with the conclusion reached by the authors that "Implementation science provides an exciting opportunity for more effectively planning, implementing, and evaluating health services programming for migrants in humanitarian contexts".

Reviewer #2: Thank you for inviting me to review this manuscript. I believe that it will make significant contributions in the field of the subject area. The article is well written. The title, abstract, methodology, results and conclusion all meet the requirements.

Reviewer #3: • Exclusion criteria: Non-English Language published materials were excluded from the scoping review.

• Table 1 should be put in Landscape form. This table is essential. It can be expanded to include type of study.

6. PLOS authors have the option to publish the peer review history of their article (what does this mean?). If published, this will include your full peer review and any attached files.

**Do you want your identity to be public for this peer review?** For information about this choice, including consent withdrawal, please see our Privacy Policy.

Reviewer #1: **Yes: **OLARINMOYE, AYODEJI OLUWADARE

Reviewer #2: **Yes: **OBIAMAKA UZOAMAKA DURUJI, PhD

Reviewer #3: **Yes: **Saheed Akinmayowa Lawal

---

## [Editor Report · Decision Letter 1]

23 Oct 2024

Innovative strategies and implementation science approaches for health delivery among migrants in humanitarian settings: a scoping review

PGPH-D-24-00863R1

Dear Dr Reynolds,

We are pleased to inform you that your manuscript 'Innovative strategies and implementation science approaches for health delivery among migrants in humanitarian settings: a scoping review' has been provisionally accepted for publication in PLOS Global Public Health.

Best regards,

Nnodimele Onuigbo Atulomah, PhD

Academic Editor

The final recommendations for corrections have been made to the satisfaction of the editor.